# A TRIZ-Adopted Development of a Compact Experimental Board for the Teaching and Learning of Operational Amplifier with Multiple Circuit Configurations

Peng Lean Chong [1,*], Silvia Ganesan [1], Poh Kiat Ng [2,*] and Feng Yuan Kong [2]

1   Department of Computer Engineering and Computer Science, School of Engineering and Computing, Manipal International University, No 1, MIU Boulevard, Putra Nilai, Nilai 71800, Malaysia
2   Faculty of Engineering and Technology, Multimedia University, Jalan Ayer Keroh Lama, Melaka 75450, Malaysia
*   Correspondence: chongpeng.lean@miu.edu.my (P.L.C.); pkng@mmu.edu.my (P.K.N.)

**Abstract:** Operational amplifiers (op-amps) are generally used for actualizing simple and complex electronic circuits in the subject of analogue electronics. In an effort to improve the teaching of op-amps in electronics engineering curricula, op-amp circuits in various configurations are often used for experiments in laboratory sessions so that students can acquire certain psychomotor and cognitive skills by constructing circuit connections and analyzing input–output waveforms. As a result, multiple configurations of operational amplifier circuits are often needed, requiring multiple sets of experimental boards or circuits for each experiment. This is usually not cost effective, requires more consumable electronic components, requires more maintenance and storage space in facilities, and is less user friendly for the students. Therefore, the aim of this research is to design a single, compact, and easy-to-replicate experimental board that can be converted into multiple configurations of the LM741 operational amplifier, comprising an inverting amplifier, a noninverting amplifier, a voltage follower, a summing amplifier, a differential amplifier, a differentiator, and an integrator, with minimal electronic components at a cost lower than EUR 10. The experimental board was tested with a constant input voltage of 1.0 V AC and a switching frequency of 1.0 kHz. It is capable of producing an output voltage corresponding to the individual operational amplifier configurations and can thus be used as a facilitating module for teaching and learning activities in the field of analogue electronics.

**Keywords:** op-amp; inverting; noninverting; voltage follower; summing; difference; differentiator; integrator

## 1. Introduction

An operational amplifier is a DC-coupled high-gain electronic voltage amplifier that has a differential input and usually a single-ended output and is designed to use components such as capacitors and resistors between its terminals [1]. The operational amplifier is often included in the curriculum of electronics studies because it can be easily and conveniently used to build simple and complex electronic amplifier circuits [2]. It amplifies the voltage difference at the input terminals with or without reference to ground and produces a voltage gain at the output, which is the product of the differential voltage and the gain [3]. Understanding how they work and their general applications is therefore fundamentally taught in almost all courses of study in electronics [4]. A well-structured way of learning operational amplifier concepts and fundamentals needs to be developed with practical psychomotor exercises, coupled with a cognitive learning curve, through theoretical instruction to avoid the problem of students completing engineering courses without being able to grasp the functional understanding of the properties of operational amplifier circuits [3].

Traditionally, strategies for teaching operational amplification in technical subjects have relied on the teaching of theoretical, practical, and experimental methods. While the first two methods can be easily integrated into any subject on operational amplifiers as the main need is the accessibility of the teaching content based on documentation or simulation, the experimental activities may prove increasingly difficult to integrate under certain circumstances. Moreover, the time constraints on conducting a particular experiment and the need for additional financial expenditures for procuring electronic components and resources for building circuits for each repetitive experiment in each semester are only additional burdens for teachers and students in each electronics course [2]. As incorporating experiments to allow students to practice hands-on skills in setting up op-amp-related circuits is important, research on developing compact and configurable op-amp experimental boards has been a focus of researchers which can be traced back to as early as 1983, when a comprehensive and programmable op-amp circuit module was developed by L. O. Kehinde [5]. This consisted of two sets of op-amps connected with various external stackable plug-in resistors and capacitors controlled by various manual mechanical switches to form five different types of op-amp circuits, namely, the inverting amplifier, noninverting amplifier, voltage follower, differentiator amplifier, and integrator amplifier. Later, in 1989, the experimental module was improved to create a single experiment board and was introduced as a dozen-impedance op-amp module [6] which could accommodate the previous types of op-amp circuits via a simplified circuit connected to stackable plug-in external resistors and capacitors without the need for mechanical switches to enable configurable connections to change from one op-amp circuit to another. However, both designs were limited in the scope of their application due to circuit design complexity which required students to understand certain logical circuitry fundamentals before they were able to understand how to use the experimental module, and they are considered relatively bulky in size compared to modern commercially available op-amp experimental boards now. With current modern technology, the development of op-amp experimental boards has shifted to the usage of complex reconfigurable hardware platforms and virtual laboratory simulators which are specifically designed to fit op-amp circuitry study such as the application of virtual instrument systems in reality (VISIR) [7,8], the MIT iLab virtual laboratory integration with the National Instruments ELVIS hardware platform [9], software simulators on Android mobile applications [10], and customized educational hardware and software kits [1,2]. Despite the current availability of simulators and extensive experimental boards that can be used in laboratories for teaching with operational amplifiers, most of them have the problem of requiring the purchase of expensive hardware and software licenses as well as annual payments for the use of software licenses with a reported estimated cost of EUR 300 up to EUR 20,000 [2], which can be an additional burden on the budget and not affordable for certain nonprofit universities, colleges, and schools from poorer backgrounds which are not supported by government funding. Thus, many institutions conduct their op-amp experiment setup via the usage of breadboards to set up each circuit configuration individually, which can be very cumbersome and time consuming and cause some problems for students because of poorly attached connections or noises that can disrupt teaching with existing time constraints [1].

Therefore, in order to effectively teach and practically demonstrate op-amps in experiments despite limited time and budgets, it is necessary that educators have access to a good and well-designed experiment board that they can incorporate into their teaching and learning strategies so that students can learn excellently. The experimental board can be used in lectures and experimental activities during laboratory sessions to develop the cognitive and psychomotor skills of students, which are essential in engineering courses [11–13] and then to strengthen their foundations in electronics.

So, the aim of this project is to design and develop a simple, compact, inexpensive, and easy-to-replicate experimental board that can be used for teaching and learning activities on the subject of operational amplifiers and that can be used in various circuit configurations covering an inverting amplifier, a noninverting amplifier, a voltage follower, a summing

amplifier, a differential amplifier, a differentiating amplifier, and an integrating amplifier for electronic subjects with minimal components at a cost lower than EUR 10. The aim is to give users the flexibility to configure different operational amplifier circuits with the same electronic components to reduce the budget for purchasing electronic components and resources for building circuits for each repetitive experiment and to facilitate teaching in a timely manner to overcome time constraints. The entire paper is organized as follows: Section 2 describes the principles of the Theory of Inventive Problem Solving (TRIZ) used in the development of the experimental board, Section 3 discusses the materials and methods used in the development of the experimental board, Section 4 describes the results of the various configurations of the operational amplifier circuits on the experimental board, and Section 5 concludes the paper.

## 2. Principles of TRIZ Applied

TRIZ is a Russian acronym for an approach known as the Theory of Inventive Problem Solving. It is a technique developed by Genrich Saulovich Altshuller (1926–1998) that provides product and process designers with inventive problem-solving tools that not only accelerate the design process but also help them to achieve world-class performance improvements that go beyond the trade-offs most designers consider as inevitable [14]. In this study, the TRIZ tools used were the cause-and-effect chain analysis (CEC), the engineering contradiction (EC), the system parameter (SP), and the inventive principle (IV). TRIZ is applicable to a wide range of inventions, disciplines, and research areas such as engineering education, design, environmental science, product development, and the service industry [15–20].

CEC analysis helps to identify the correct cause(s) of the problem. If the wrong cause is identified, it will lead to an unsuccessful response [21]. In essence, CEC analysis is similar to the "5 Whys" process of continuously eliciting the causes of the problem by asking "why", moving from high-level causes to low-level causes [22]. CEC analysis has proven to be one of the more popular tools for a number of reasons, including the simplicity of its principles, its ability to be applied to a wide range of issues with varying degrees of complexity, its ability to drill down to the level of atoms if necessary, and the ease with which its findings can be shared [23]. Based on the findings from previous studies, a simple CEC analysis can be carried out. Figure 1 shows the CEC analysis performed for the main problem.

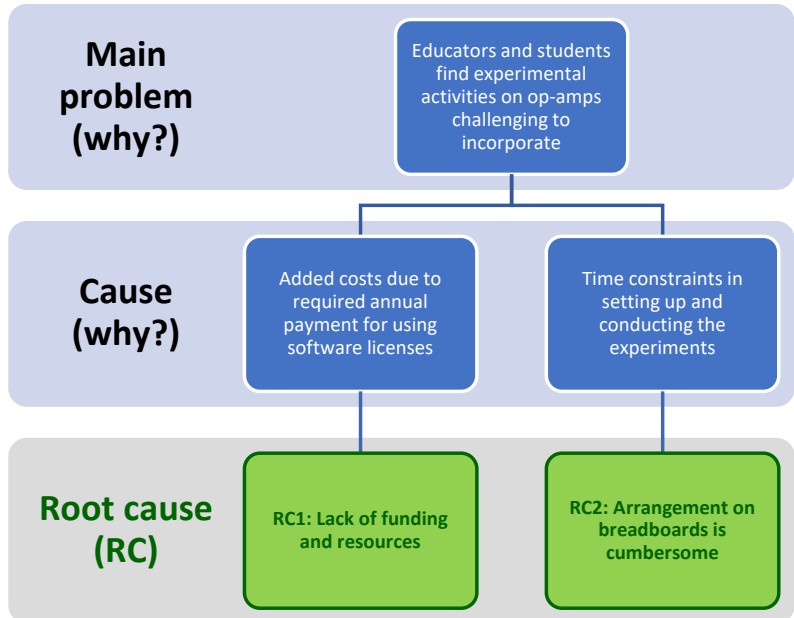

**Figure 1.** Cause-and-effect chain (CEC) analysis.

The two root causes for the main problem are: (RC1) lack of funding and resources; and (RC2) the arrangement on breadboards is cumbersome. According to the identified causes, two engineering contradiction statements can be formulated as follows:

EC1 (based on RC1): if many high-quality simulators and experimental boards are purchased for the operational amplifier course, then it is less difficult to incorporate the experimental activities into the course, but the cost of maintaining the resources is very high.

EC2 (based on RC2): if technicians or lab assistants are hired to help with the tedious arrangements on the breadboards, then the breadboard arrangement can be undertaken in a short time, but more resources and expenditure on staff are required.

The "then" and "but" parts of the ECs are then linked to the 39 system parameters of TRIZ. The 39 system parameters are established and fixed general characteristics of a system [24]. The contradiction matrix is made up of the 39 system parameters that describe any engineering system [25]. If one desires to increase one parameter, the other parameter associated with it immediately deteriorates. The engineering contradiction at the heart of the problem is created by the correlation of the two parameters [26]. The parameters are then crossed in the contradiction matrix to identify the most appropriate inventive principle(s) to be chosen to guide the resolution of the contradiction.

The examination of 40,000 patent inventions produced the 40 inventive principles, which give designers guidance in selecting the best solution for a design challenge [27]. Table 1 shows the summary of the inventive principles identified from the intersection of the selected TRIZ system parameters (SPs) in the contradiction matrix. In the end, the universality principle (IV #6) was selected to solve EC1, while the principle of another dimension (IV #17) was selected to solve EC2.

**Table 1.** Summary of inventive principles identified from intersection of system parameters.

| Details | EC1 | EC2 |
|---|---|---|
| "then" part (with system parameter) | then it is less difficult to incorporate the experimental activities into the course (SP #35 Versatility) | then the breadboard arrangement can be undertaken in a short time (SP #15 Duration of action by moving object) |
| "but" part (with system parameter, SP) | but the cost of maintaining the resources is very high (SP #39 Productivity) | but more resources and expenditure on staff are required (SP #39 Productivity) |
| Inventive principle (IV) | IV #35: Parameter changes IV #28: Mechanics substitution IV #6: Universality IV #37: Thermal expansion | IV #35: Parameter changes IV #17: Another dimension IV #14: Curvature IV #19: Periodic action |
| Selected | IV #6: Universality | IV #17: Another dimension |

**Universality:** In TRIZ, the universality principle refers to the approach of allowing a part or object to perform multiple functions [28]. The universality principle has been used in research pertaining to food packaging, hoverboard design, service design, tracked robot design, and building design [29–33]. Using this principle, researchers propose to design the PCB in such a way that it is able to support multiple functions on a single board, such as an inverting amplifier, a noninverting amplifier, a voltage follower, a summing amplifier, a differential amplifier, a differentiating amplifier, and an integrator amplifier. This solution solves EC1 by eliminating the need to spend more money on maintaining multiple resources. According to Lim and Ng [34], systems and tools that are less useful may be eliminated over time, while the more universal and multifunctional ones will continue to be used.

**Another dimension:** The principle of another dimension refers to moving an object from a one- or linear-dimensional space into a multidimensional space [35]. It is basi-

cally a realignment of an object. The principle of another dimension has been used in studies pertaining to crowd management, design optimization, interaction design, and vehicle dynamics [36–39]. With this principle, instead of a single circuit configuration on a single board, the PCB could be adapted to support multiple circuit configurations by optimizing the specification and positioning of the electronic components. This solution allows flexibility in modifying the circuit to accommodate or support multiple operational amplifier configurations.

### 3. Material and Methods

The syllabus for teaching operational amplifiers in electrical engineering usually covers the basic circuit configurations such as inverting amplifiers, noninverting amplifiers, voltage followers, summing amplifiers, differential amplifiers, differentiating amplifiers, and integrating amplifiers [40–44]. In order to accommodate the different circuit configurations on a single PCB, optimization of the specification and positioning of the electronic components is critical to allow flexibility of circuit changes to accommodate and support multiple operational amplifier configurations. Therefore, a schematic of the operational amplifier experimental board with multiple circuit configurations was designed and is shown in Figure 2. Each of the circuit configurations is discussed here piece by piece. A single feedback resistor $R_f$ and an input resistor $R_1$ make up the inverting amplifier circuit with negative feedback network shown in Figure 2a. The positive input terminal of the op-amp is grounded, and the positive input terminal of the power supply is connected to the negative input terminal of the op-amp. The output terminal will amplify the input voltage while inverting its polarity. Through the external feedback resistor $R_f$, the output voltage is continually fed back to the inverting input terminal. As a result, the input voltage and the amplified output voltage can be stated as follows [40]:

$$V_{out} = -\left(\frac{R_f}{R_1}\right)(V_{in})$$ (1)

In the noninverting amplifier in Figure 2b, feedback control of the noninverting operational amplifier is achieved by amplifying a small portion of the output voltage signal back to the noninverting input terminal by means of a feedback resistor $R_f$, without polarity reversal at the output. The positive input of the power supply is connected to the positive input terminal of the operational amplifier, while the negative input terminal of the operational amplifier is grounded. Therefore, the output voltage has the same polarity but is higher than the input voltage, as shown in the formula below [40]:

$$V_{out} = \left(\frac{R_f}{R_1} + 1\right)(V_{in})$$ (2)

In the voltage follower circuit in Figure 2c, the output voltage follows the same magnitude and polarity as the input voltage, with no external resistors involved. The voltage follower is often used as a buffer in circuit design. Therefore, the output voltage signal that exactly follows the input voltage signal is expressed as follows [40]:

$$V_{out} = V_{in}$$ (3)

In the inverting summing amplifier shown in Figure 2d, the circuit is designed to take up to 3 input signals and perform the addition of input voltages $V_1$, $V_2$, and $V_3$, with the input sources connected to the inverting input terminal of the operational amplifier, while the positive input terminal of the operational amplifier is grounded. It is often used as a signal mixer to combine audio signals from different input sources such as microphones, electric musical instruments, and others into a single combined output signal. Therefore, the output voltage signal of the inverting summing amplifier is proportional to the algebraic sum of its individual inputs with reversed polarity. The output voltage signal is given by [40]:

$$V_{\text{out}} = -R_f \left( \frac{V_1}{R_1} + \frac{V_2}{R_2} + \frac{V_3}{R_3} \right) \tag{4}$$

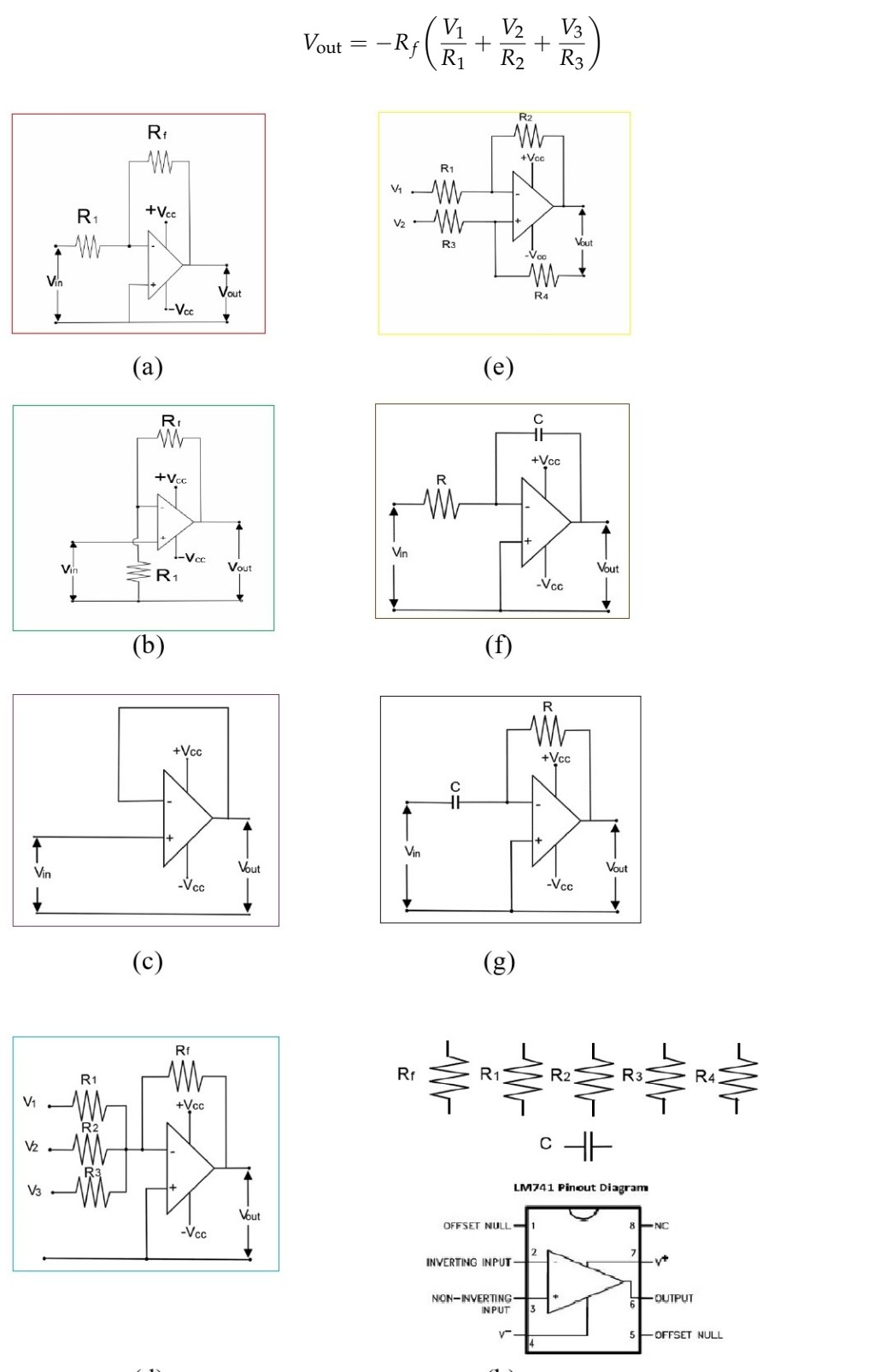

**Figure 2.** Schematic diagram of the experimental board for op-amp. An inverting amplifier is shown in (**a**), an non-inverting amplifier in (**b**), a voltage follower in (**c**), a summing amplifier in (**d**), a difference amplifier in (**e**), an integrator amplifier in (**f**), a differentiator amplifier in (**g**), and the allocation of the electronic components positioning is in (**h**).

In the case where the feedback resistor $R_f$ is designed to have the same magnitude as the input resistors $R_1$, $R_2$, and $R_3$, where $R_1 = R_2 = R_3 = R_f$, the formula in (4) can be simplified as:

$$V_{\text{out}} = -(V_1 + V_2 + V_3) \tag{5}$$

Next, the inverting differential amplifier shown in Figure 2e detects the voltage difference at its inverting and noninverting input terminals, which is amplified at the output terminal of the operational amplifier. The input terminal can optionally be grounded or connected to the input voltage sources to perform the subtraction process. Therefore, the output voltage $V_{\text{out}}$ is proportional to the voltage difference in the input terminals, as shown below [40]:

$$V_{\text{out}} = \left( \frac{R_3(R_1 + R_2)}{R_1(R_3 + R_4)} \right) V_2 - \left( \frac{R_2}{R_1} \right) V_1 \tag{6}$$

For the condition that the input resistors are designed as $R_1 = R_4$ and $R_2 = R_3$, the formula in (6) can be simplified as:

$$V_{\text{out}} = \left( \frac{R_2}{R_1} \right) (V_2 - V_1) \tag{7}$$

The inverting integrator amplifier shown in Figure 2f performs the mathematical operation of integration, which allows the output to respond to changes in the input voltage over time. The integrator amplifier produces an output voltage that is proportional to both the integral of the amplitude of the input voltage and the duration of the input signal. The magnitude of the output signal is thus determined by the duration (from zero seconds to $t$ seconds) that a voltage is applied to its input terminals, as the current through the feedback loop charges or discharges the capacitor as the required negative feedback is provided by the capacitor. Resistor $R_1$ is connected to the input terminal of the inverting amplifier, while capacitor $C_f$ provides the negative feedback through the operational amplifier, and the positive terminal is grounded to produce an inverting output voltage $V_{\text{out}}$, which can be expressed as follows [40]:

$$V_{\text{out}} = -\frac{1}{R_1 C_f} \int_0^t V_i(\tau) dt \tag{8}$$

For simplification, the formula in (8) can also be rewritten as follows:

$$V_{\text{out}} = -\frac{1}{j\omega R_1 C_f} V_i \tag{9}$$

The final circuit configuration is the inverting differential amplifier in Figure 2g. It generates the first derivative of the input signal. Compared to the integrator amplifier, the positions of the capacitor and the resistor have been swapped. This results in the capacitor $C_1$ being connected to the input terminal of the inverting amplifier, while the resistor $R_f$ forms the negative feedback element across the operational amplifier, with the positive terminal grounded, to produce an inverting output voltage $V_{\text{out}}$, which can be expressed as follows [40]:

$$V_{\text{out}} = -R_f C_1 \frac{dV_i}{dt} \tag{10}$$

Figure 2h shows the arrangement of the electronic passive components covering several resistor slots and one capacitor slot, as well as an LM741 operational amplifier used as the body component for all of the circuit configurations for the operational amplifier shown in Figure 2a–g. The entire circuit configurations used the same set of electronic components on the experimental board, where each of the component slots can be populated with different values of resistors and capacitors depending on user preference. For verification purposes in this project, each of the electronic components is predefined in order to be able to test the experimental board.

The entire printed circuit board (PCB) design of the experimental board is arranged as shown in Figure 3, which portrays the design of the board with the op-amp circuit configurations on one side and the slot for the electronic components used to build the circuit configuration on the other. It shows that only a set of electronic components such as the LM741 DC-coupled high gain electronic voltage amplifier, a capacitor labeled C, and 5 resistors labeled $R_f$, $R_1$, $R_2$, $R_3$, $R_4$ are sufficient to be used to construct the 7 different types of op-amp configuration covering the inverting amplifier, noninverting amplifier, voltage follower, summing amplifier, difference amplifier, integrator amplifier, and differentiator amplifier with a total material cost lower than EUR 10. The full design of the PCB is shown here to enable easy replication by any interested instructors or students to fabricate the experiment board to support teaching and learning of related op-amp configuration circuits in their subject laboratory sessions. The circuits can be connected using jumper wires accordingly to connect the electronic components towards any of the selected op-amp circuits, and the user can build the desired circuit with practical exercises to develop their psychomotor and cognitive skills.

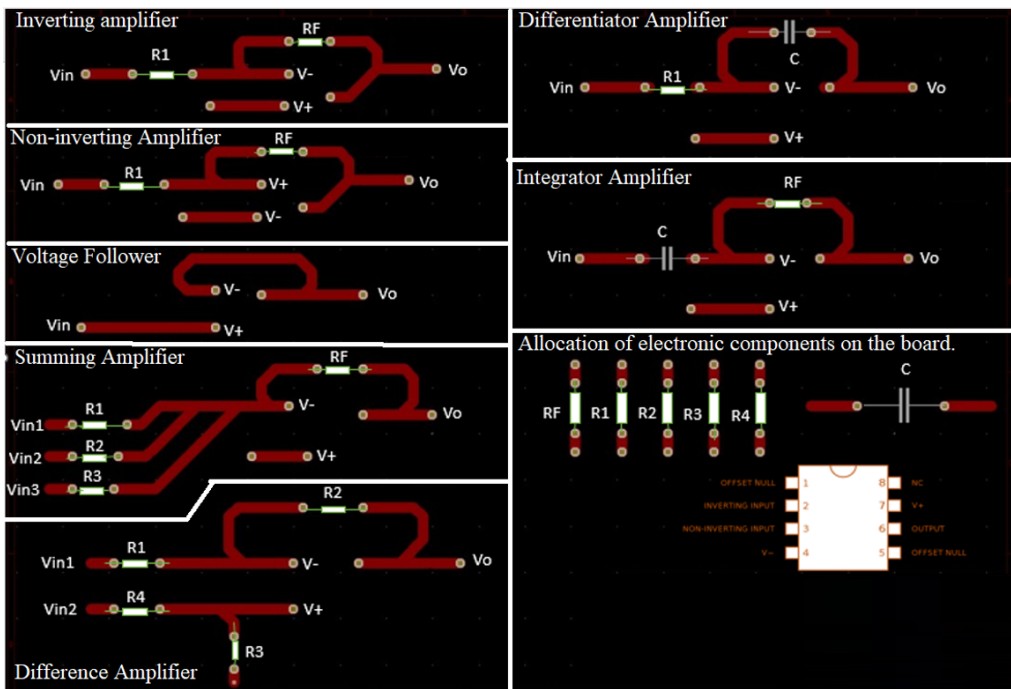

**Figure 3.** Printed circuit board layout for the replicable experimental board.

A real fabricated experimental board was made, as shown in Figure 4, based on the PCB design in Figure 3 which shows each of the operational amplifier circuit configurations. The circuit board can be easily fabricated via the normal "do it yourself" (DIY) fabrication process for PCBs at home or in the lab. All of the op-amp configurations were tested and proven to be functionable by setting them up via a plug-and-play fashion to connect to the same set of electronic components.

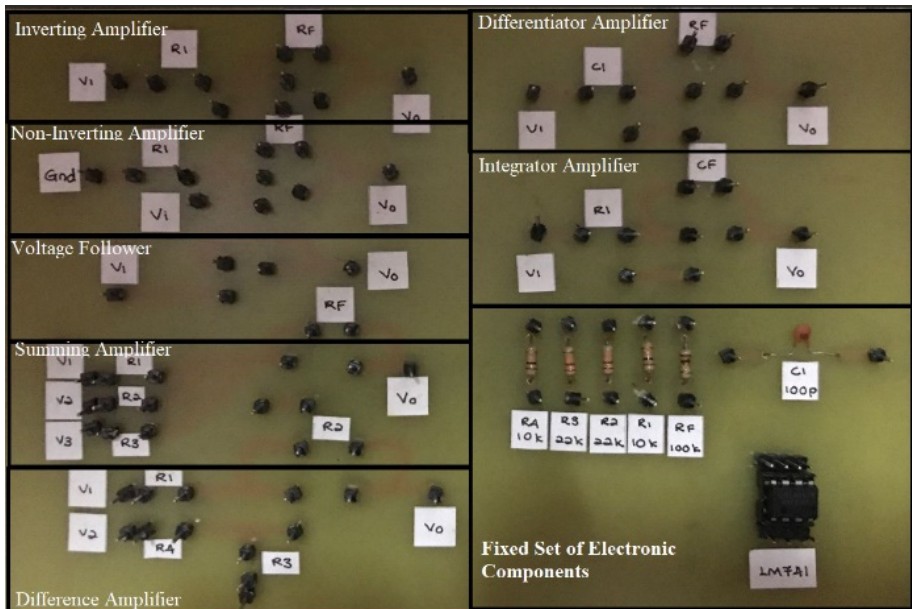

**Figure 4.** A portable and compact experimental board for teaching and learning of operational amplifier with multiple circuit configurations.

The procedure to show how to use the experimental board and how to construct the desired operational amplifier circuit is shown in the flowchart in Figure 5 to guide students and enable them to follow the procedure systematically to complete any op-amp circuit configuration with ease. The step-by-step guidance can reduce the circuit construction complexity and provide them with clear instructions on how to use the experimental board.

First, the student must select the type of operational amplifier circuit they wish to build. Next, the resistors or capacitor values must be selected from the given range of resistors $R_f$, $R_1$, $R_2$, $R_3$, and $R_4$ and capacitor $C_f$, according to the instructions provided with the board. Then the pins of the LM741 op-amp must be connected to the pins of the selected op-amp circuit via plug-and-play jumper wires. The same jumpers should also be connected to the selected resistors or capacitors to match the previously selected op-amp circuit. Once connected, the student can connect the input voltage to the $+V_{cc}$ and $-V_{cc}$ pins of the op-amp to start the op-amp and connect the $V_{in}$ input pins of the selected op-amp configuration circuit to the function generator to generate the input signals. On the other hand, the student can connect the $V_o$ output pins to the oscilloscope to capture the output waveform and the multimeter to all the pins needed for signal measurement. The data result can be recorded for analysis, and once this is completed, all the jumper connections can be removed. The connections can be repeated for all 7 operational amplifier configuration circuits, i.e., the inverting amplifier, the noninverting amplifier, the voltage follower, the summing amplifier, the differential amplifier, the differentiator, and the integrator circuits in a similar way with the same LM741 operational amplifier, resistors, and capacitors. If an electronic component breaks and needs to be replaced, an educator can simply desolder it and replace it with a new component that he/she plugs into the pin headers provided on the board.

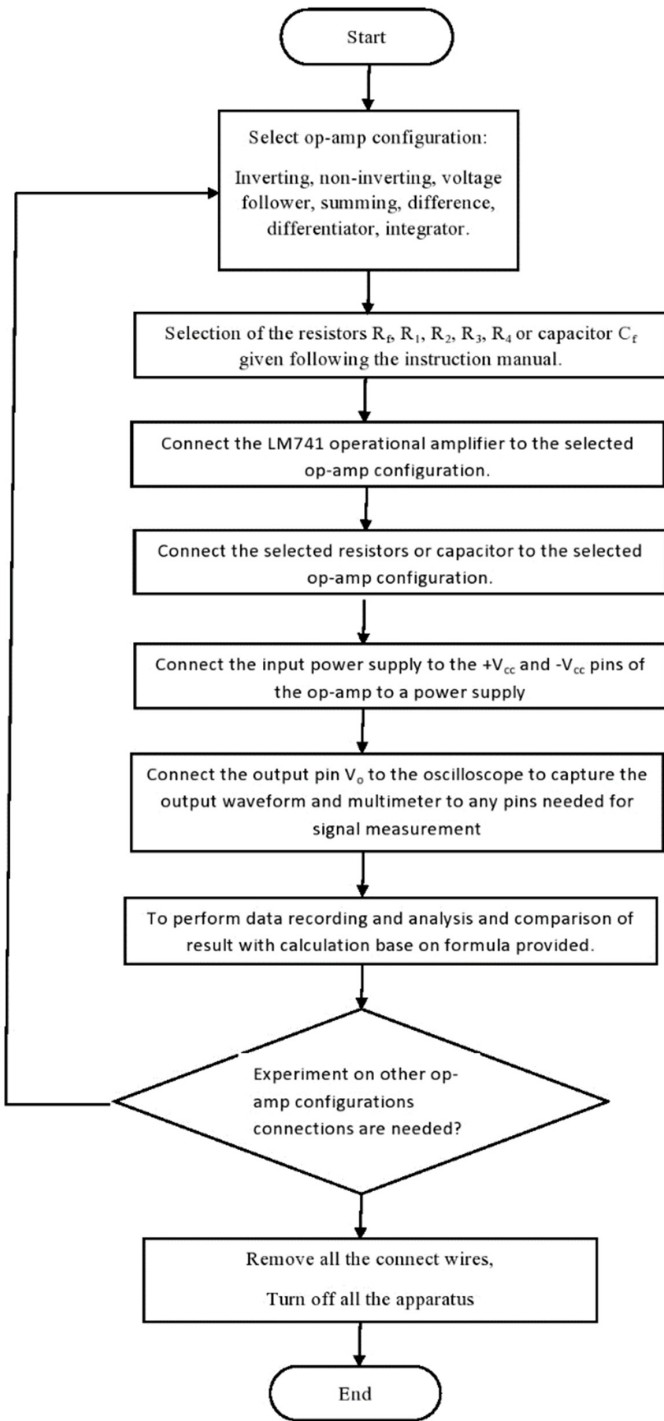

**Figure 5.** Flowchart on the procedure to utilize the experimental board.

## 4. Result and Discussion

An analysis of the input voltage $V_{in}$ and the output voltage $V_o$ of the operational amplifier was performed to verify the performance of the experimental board for the various operational amplifier circuit configurations, which included inverting amplifiers, noninverting amplifiers, voltage followers, summing amplifiers, differential amplifiers, differentiators, and integrator circuits. Each of the waveform analyses for each configuration is shown in Figure 6, which shows a comparison of the input waveform and output waveform during the testing phase to determine the complete functionality of the experimental board.

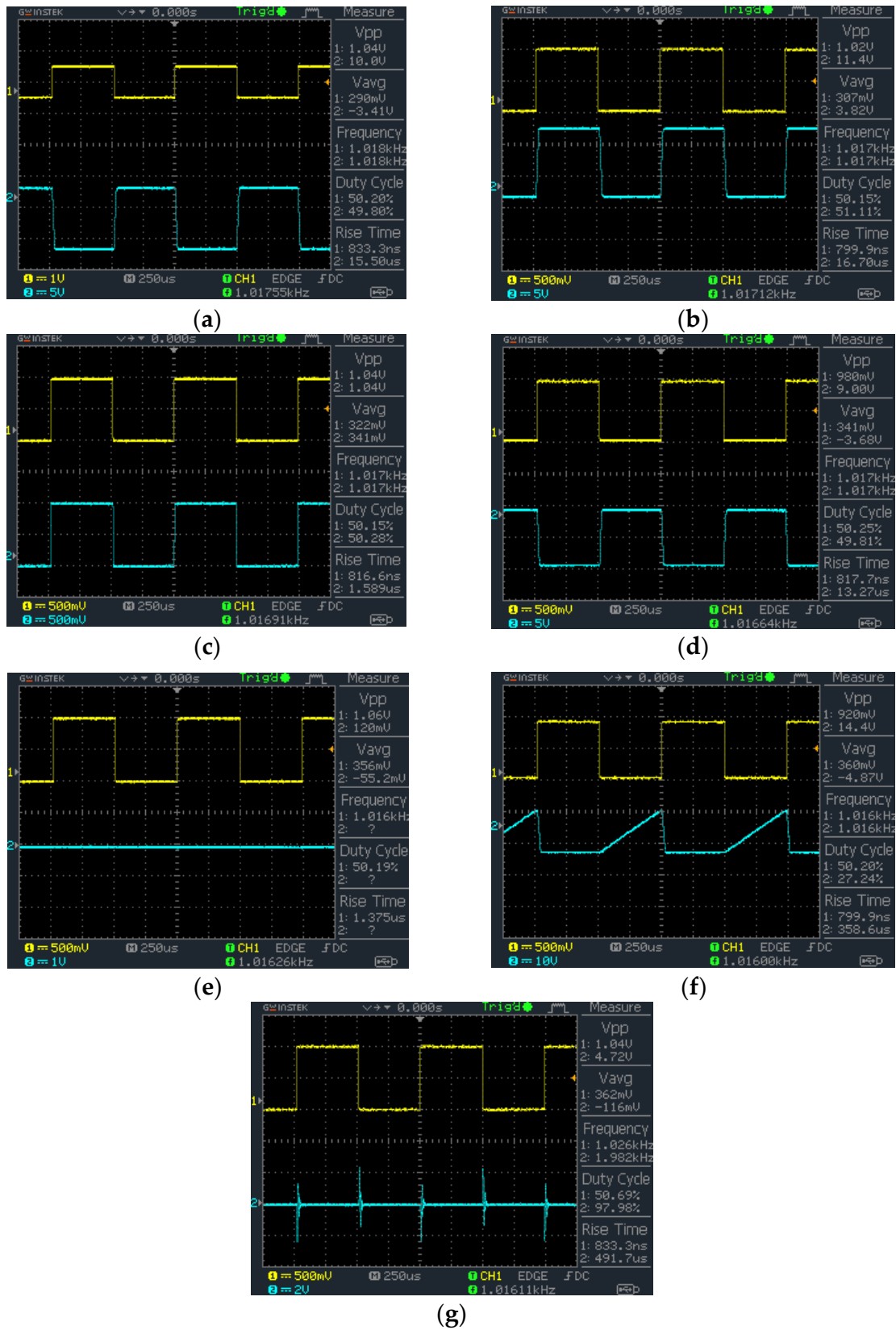

**Figure 6.** Input waveform and output waveform of all of the op-amp configurations on the experimental board. The waveform of inverting amplifier is shown in (**a**), noninverting amplifier in (**b**), voltage follower in (**c**), summing amplifier in (**d**), difference amplifier in (**e**), integrator amplifier in (**f**), and differentiator amplifier in (**g**).

Figure 6a shows the result of the inverting amplifier. Resistors $R_f$ = 100 kΩ and $R_1$ = 10 kΩ were inserted into the $R_f$ and $R_1$ slots. A voltage of 10 V DC was applied to the $+V_{cc}$ and $-V_{cc}$ pins of the LM741 op-amp to activate the op-amp. Then, a square wave with a frequency of 1 kHz and a peak-to-peak voltage of 1.04 V ($V_{pp,in}$) was applied to the input pin $V_{in}$. After amplification, the result shows that the waveform was amplified from $V_{pp,in}$ = 1.04 V to $V_{pp,out}$ = 10.0 V with a gain $A_v$ of about 10 according to the formula in (1). In addition, the input and output waveforms were inverted due to the construction of the inverting amplifier operational amplifier.

The result of the noninverting amplifier can be seen in Figure 6b. The same $R_f$ and $R_1$ were used. In addition, the same $+V_{cc}$ and $-V_{cc}$ were applied to activate the operational amplifier. A 1 kHz frequency with a 1.02 $V_{pp,in}$ square wave was applied to the input pin, $V_{in}$. After amplification, the result in the diagram shows that the waveform was amplified from $V_{pp,in}$ = 1.02 V to $V_{pp,out}$ = 11.4 V, which corresponds to an amplification of $A_v$ equal to 11, as indicated in the formula in (2). Both the input and output waveforms retained the same polarity due to the noninverting circuitry of the operational amplifier.

In addition, Figure 6 shows the result of the voltage follower amplifier. The same amount of $+V_{cc}$ and $-V_{cc}$ was applied to activate the op-amp, but no resistors were needed due to the nature of the voltage follower amplifier connection. A 1 kHz frequency of 1.04 $V_{pp,in}$ in the form of a square wave was applied to the input pin $V_{in}$. Therefore, both $V_{in}$ and $V_{out}$ were equal to 1.04 V with $A_v$ = 1, which corresponds to the formula in (3), which states that both the input and output voltages are equal because the voltage follower operational amplifier only acts as a buffer.

Furthermore, Figure 6d shows the result of the summing amplifier. The same magnitude of $+V_{cc}$ and $-V_{cc}$ was used to drive the operational amplifier. The resistors $R_f$ = 100 kΩ, $R_1$ = 10 kΩ, $R_2$ = 22 kΩ, and $R_3$ = 22 kΩ were chosen accordingly. A frequency of 1 kHz with about 0.98 $V_{pp}$ in the form of a square wave was simultaneously connected to all three input pins, namely, $V_{in1}$, $V_{in2}$, and $V_{in3}$. The output waveform showed a value of $V_{pp,out}$ = 9.0 V with inverted polarity because $V_1$, $V_2$, and $V_3$ were connected to the inverting input $-V_{in}$, while $+V_{in}$ was grounded. The result is confirmed by the formula in (4), which shows the gain of the values of $V_{pp,in}$ to $V_{pp,out}$ with the selected resistors $R_f$, $R_1$, $R_2$, and $R_3$.

In addition, the result of the differential amplifier is shown in Figure 6e. To activate the operational amplifier, $+V_{cc}$ and $-V_{cc}$ were applied in the same order of magnitude. The resistors chosen were $R_f$ = 100 kΩ, $R_1$ = 10 kΩ, $R_2$ = 22 kΩ, $R_3$ = 22 kΩ, and $R_4$ = 10 kΩ. A frequency of 1 kHz with 1.06 $V_{pp}$ in the form of a square wave was applied simultaneously to the two input pins $V_{in1}$ and $V_{in2}$. After the difference was formed by the operational amplifier, it was observed that the output waveform was a straight line with $V_{pp,out}$ = 0.12 V, which was approximately zero voltage. According to Formula (7), the output voltage $V_{pp,out}$ = 0.12 V is 0 V when the two inputs $V_{in1}$ and $V_{in2}$ have the same value, and the resistors $R_1$ = $R_4$ and $R_2$ = $R_3$.

Next, Figure 6f shows that the result of the integrator amplifier. The same magnitude of $+V_{cc}$ and $-V_{cc}$ was applied to activate the operational amplifier. The selected resistor was $R_1$ = 100 kΩ, and the capacitor $C_f$ = 100 pF. A frequency of 1 kHz with 0.92 $V_{pp}$ in the form of a square wave was applied to the input pin $V_{in}$. The sawtooth-shaped output waveform showed a value of $V_{pp,out}$ = 14.4 V, whereby the waveform corresponds to expectations and agrees with the calculated result according to Formula (9).

Also, Figure 6g shows the result of the differential amplifier. The same amount of $+V_{cc}$ and $-V_{cc}$ was used to activate the operational amplifier. The selected resistor was $R_f$ = 100 kΩ, and the capacitor $C_1$ = 100 pF. A frequency of 1 kHz with 1.04 $V_{pp}$ in the form of a square wave was applied to the input pin $V_{in}$. The peak-shaped output waveform showed a value of $V_{pp,out}$ = 4.72 V, with the waveform matching expectations and the calculated result matching the formula in (10). A summary of the calculations involving the formulae used and parameters involved for each op-amp configurations is shown in Table 2.

**Table 2.** Summary of formulae and parameters used for each op-amp configurations.

| Number | Op-amp Circuit Types | Formulae | Input Voltage, $V_{in}$ | Resistors, $R$ | Capacitors, $C$ | Output Voltage, $V_{out}$ |
|---|---|---|---|---|---|---|
| 1 | Inverting amplifier | $V_{out} = -\left(\frac{R_f}{R_1}\right)(V_{in})$ | 1.04 V | $R_f = 100\ k\Omega$, $R_1 = 10\ k\Omega$ | - | 10.0 V |
| 2 | Noninverting amplifier | $V_{out} = \left(\frac{R_f}{R_1} + 1\right)(V_{in})$ | 1.02 V | $R_f = 100\ k\Omega$, $R_1 = 10\ k\Omega$ | - | 11.40 V |
| 3 | Voltage follower | $V_{out} = V_{in}$ | 1.04 V | - | - | 1.04 V |
| 4 | Summing amplifier | $V_{out} = -R_f\left(\frac{V_1}{R_1} + \frac{V_2}{R_2} + \frac{V_3}{R_3}\right)$ | 0.98 V | $R_f = 100\ k\Omega$, $R_1 = 10\ k\Omega$, $R_2 = 22\ k\Omega$, $R_3 = 22\ k\Omega$ | - | 9.00 V |
| 5 | Differential amplifier | $V_{out} = \left(\frac{R_2}{R_1}\right)(V_2 - V_1)$ | 1.06 V | $R_f = 100\ k\Omega$, $R_1 = 10\ k\Omega$ $R_2 = 22\ k\Omega$, $R_3 = 22\ k\Omega$, $R_4 = 10\ k\Omega$ | - | 0.12 V |
| 6 | Integrating amplifier | $V_{out} = -\frac{1}{j\omega R_1 C_f} V_i$ | 0.92 V | $R_1 = 100\ k\Omega$ | $C_f = 100\ pF$ | 14.40 V |
| 7 | Differentiating amplifier | $V_{out} = -R_f C_1 \frac{dV_i}{dt}$ | 1.04 V | $R_f = 100\ k\Omega$ | $C_1 = 100\ pF$ | 4.72 V |

## 5. Conclusions

This paper proposes a TRIZ-adopted compact experimental board consisting of several configurations of operational amplifiers, including an inverting amplifier, a noninverting amplifier, a voltage follower, a summing amplifier, a differential amplifier, a differentiating amplifier and an integrating amplifier suitable for teaching and learning on the subject of operational amplifiers at a cost lower than EUR 10. Compared to the commercially available experimental boards or virtual simulators with an estimated price range from EUR 300 up to EUR 20,000 [2], this proposed experimental board which costs less than EUR 10 and is easy to replicate by instructors or students, via a DIY manner, to be accommodated into their experiment sessions, is indeed providing a huge cost saving to the users. Instead of constructing each of the seven op-amp configuration individually on breadboards with different sets of electronic components, the usage of this experimental board proposes a quick and easy-to-use system for the students where they can set up the circuit with the same set of electronic components at an estimation of 85% time saving. The design of the board is such that all the different circuit configurations can be created with the same set of electronic components. The user only needs to connect the predefined pins for each configuration to the fixed electronic components and execute the desired function. Each circuit configuration has been experimentally checked for its input and output waveforms and matched with the appropriate formulae. The idea of the proposed design aims to develop a low-cost, compact, portable, and easy-to-use experimental board suitable for the study and analysis of operational amplifier circuits, which are considered compulsory in most electrical engineering subjects. The proposed board can simplify the conduct of experiments by reducing the complexity of experiments for students, enabling faster circuit setup time, and reducing the cost of housing several different boards for different operational amplifier configurations. It requires fewer consumable electronic components, requires less maintenance work to sustain the usage of the board, and needs less storage space in institutions when compared to the traditional setup methods of using multiple breadboards and electronic components to set up similar circuit configurations. The proposed experimental board can be used in experimental laboratory exercises with operational amplifier circuits for analogue electronics courses in primary, secondary, and higher education.

**Author Contributions:** Conceptualization, P.L.C. and P.K.N.; methodology, P.L.C. and S.G.; software, S.G.; validation, P.L.C. and S.G.; formal analysis, P.L.C., P.K.N., S.G., and F.Y.K.; investigation, P.L.C. and S.G.; resources, P.L.C., P.K.N., and S.G.; data curation, P.L.C., S.G., and F.Y.K.; writing—original draft preparation, P.L.C., P.K.N., S.G., and F.Y.K.; writing—review and editing, P.L.C. and P.K.N.; visualization, P.L.C., P.K.N., S.G., and F.Y.K.; supervision, P.L.C.; project administration, P.L.C.; funding acquisition, P.L.C. and P.K.N. All authors have read and agreed to the published version of the manuscript.

**Funding:** This research received no external funding.

**Institutional Review Board Statement:** Not applicable.

**Informed Consent Statement:** Not applicable.

**Data Availability Statement:** Not applicable.

**Acknowledgments:** The researchers would like to thank the Center of Sustainable Communication and the IoT (CSCIoT) of Multimedia University for providing the lab and facility for this project throughout the research.

**Conflicts of Interest:** The authors declare no conflict of interest.

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
