# Peer review of "A TRIZ-Adopted Development of a Compact Experimental Board for the Teaching and Learning of Operational Amplifier with Multiple Circuit Configurations"

_sustainability, doi:10.3390/su142114115_

Round 1

Reviewer 1 Report

The article "A TRIZ-Adopted Development of a Compact Experimental Board for Teaching and Learning of Operational Amplifier with Multiple Circuit Configurations” are interesting.

However, it would be even better if the following issues are considered: 

1. 
 References are limited and, in some cases, the context from the specified reference text is unclear.

2.  Add more details on methods and materials that are required and need a reference to the statistical processing of data. Several experiments (repetitions) and statistics of the results are not straightforward.

3. FIG 3-6, Small, unclear. Necessity and contribution must be explained

4.    Lacks a detailed explanation of the calculation formulas and parameters

5. Extending the conclusions – what are the additional topics that are required for review? 

Author Response

Dear respected reviewer;

Please see the attachment for my response to each of your kind comments. 

Thank you. Appreciated.

Reviewer 2 Report

This manuscript proposed a single compact experimental board that can be used for teaching and learning activities on the subject of operational amplifiers and that can be converted into multiple circuit configurations for electronic subjects with minimal electronic components. The flow is clear and intact. However, some corrections and additions are required.

1.    In figure 2h, the text is unclear.

2.    The authors are suggested to move the Figure 3 down a little bit. So, there will be enough space between the text and figure.

3.    It will be great if the authors can more details about Figure 3? Also, the text on the downright conner is not clear as well.

4.    Figure 3 is not aligned with the other figures in the manuscript.

5.    Figure 6 is unclear. The resolution should be improved, otherwise, those figures cannot give that much information.

6.    Is there any other novelty except saving the budget and time in the classroom?

7.    Can you estimate how much money and time the proposed device can save compared to the traditional or commercially available boards?

8.    Is there any other paper that also contributes to the similar area? In the introduction section, I did not see enough background investigation regarding to the progress of research in this area.

Author Response

(The authors gave the same response as above.)

Round 2

Reviewer 1 Report

There was a reference to the comments I gave. The article has improved. can be accepted.